# Differential Diagnosis of Alzheimer Disease vs. Mild Cognitive Impairment Based on Left Temporal Lateral Lobe Hypomethabolism on ^18^F-FDG PET/CT and Automated Classifiers

**DOI:** 10.3390/diagnostics12102425

**Published:** 2022-10-07

**Authors:** Susanna Nuvoli, Francesco Bianconi, Maria Rondini, Achille Lazzarato, Andrea Marongiu, Mario Luca Fravolini, Silvia Cascianelli, Serena Amici, Luca Filippi, Angela Spanu, Barbara Palumbo

**Affiliations:** 1Unit of Nuclear Medicine, Department of Medicine, Surgery and Pharmacy, Università degli Studi di Sassari, 07100 Sassari, Italy; 2Department of Engineering, Università degli Studi di Perugia, 06125 Perugia, Italy; 3Cognitive Disorders and Dementia Unit, USL Umbria 1, 06127 Perugia, Italy; 4Department of Nuclear Medicine, Santa Maria Goretti Hospital, 04100 Latina, Italy; 5Section of Nuclear Medicine and Health Physics, Department of Medicine and Surgery, Università degli Studi di Perugia, 06132 Perugia, Italy

**Keywords:** Alzheimer disease, mild cognitive impairment, brain ^18^F-FDG PET/CT, artificial intelligence, automatic classification

## Abstract

Purpose: We evaluate the ability of Artificial Intelligence with automatic classification methods applied to semi-quantitative data from brain ^18^F-FDG PET/CT to improve the differential diagnosis between Alzheimer Disease (AD) and Mild Cognitive Impairment (MCI). Procedures: We retrospectively analyzed a total of 150 consecutive patients who underwent diagnostic evaluation for suspected AD (n = 67) or MCI (n = 83). All patients received brain 18F-FDG PET/CT according to the international guidelines, and images were analyzed both Qualitatively (QL) and Quantitatively (QN), the latter by a fully automated post-processing software that produced a z score metabolic map of 25 anatomically different cortical regions. A subset of n = 122 cases with a confirmed diagnosis of AD (n = 53) or MDI (n = 69) by 18–24-month clinical follow-up was finally included in the study. Univariate analysis and three automated classification models (classification tree –ClT-, ridge classifier –RC- and linear Support Vector Machine –lSVM-) were considered to estimate the ability of the z scores to discriminate between AD and MCI cases in. Results: The univariate analysis returned 14 areas where the z scores were significantly different between AD and MCI groups, and the classification accuracy ranged between 74.59% and 76.23%, with ClT and RC providing the best results. The best classification strategy consisted of one single split with a cut-off value of ≈ −2.0 on the z score from temporal lateral left area: cases below this threshold were classified as AD and those above the threshold as MCI. Conclusions: Our findings confirm the usefulness of brain 18F-FDG PET/CT QL and QN analyses in differentiating AD from MCI. Moreover, the combined use of automated classifications models can improve the diagnostic process since its use allows identification of a specific hypometabolic area involved in AD cases in respect to MCI. This data improves the traditional 18F-FDG PET/CT image interpretation and the diagnostic assessment of cognitive disorders.

## 1. Introduction

Alzheimer disease (AD), the most frequent form of neurodegenerative dementia [1,2], has an increasing incidence due to the progressive aging of the population [3]. The disease evolution of AD is a progressive continuum starting from the subclinical phase of Mild Cognitive Impairment (MCI), which is characterized by the absence of objective evidence of damage to functional autonomy [4,5]. This condition is considered high risk for AD [6]: approximately 10% of MCI cases per year progress to AD or other forms of dementia; however, a fraction of MCI patients will not develop clinical dementia, even after 10 years [5,7,8].

It is, therefore, crucial to identify those MCI cases that are more likely to progress to AD or other forms of dementia. This would allow direct patients toward adequate clinical trials or prevention strategies since no disease-modifying therapy is currently available [1,9].

There are different tools to evaluate the risk of conversion to AD; among them, the assessment of cerebral metabolism by ^18^F-fluoro-deoxyglucose-PET/CT (^18^F-FDG PET/CT) either alone or in conjunction with other procedures is one of the most accurate [10,11,12,13,14,15]. Moreover, this procedure can be strengthened by semi-quantitative evaluation [16,17], which provides reproducible, standardized parameters.

In recent years, Artificial Intelligence (AI) methods—including machine learning, deep learning and radiomics—have been successfully applied to neurological diseases, particularly to contribute to the diagnosis of Parkinson’s disease and dementia [18,19,20]. Various authors have advocated the use of automatic classification for discriminating between AD and MCI [21] based on MR images [22,23,24] or ^18^F-FDG PET/CT images [21,25,26], also combined with different types of biomarkers.

The aim of our study was to evaluate further the ability of AI applied to semi-quantitative data from ^18^F-FDG PET/CT of the brain to improve the diagnosis of cognitive disorders and, in particular, AD and MCI.

## 2. Materials and Methods

### 2.1. Study Population

We retrospectively investigated 150 consecutive patients evaluated for cognitive impairment (64 males, 86 females; age = 70.59 ± 9.14 [40–85] year) who underwent ^18^F-FDG brain PET for differential diagnosis of MCI and dementia between November 2017 and January 2021. Of the 150 cases, 67 had suspected AD (29 males and 38 females) and 83 suspected MCI (35 males and 48 females).

All patients were evaluated for neurological familiar diseases and neurological- and general-related diseases (hypertension and diabetes mellitus). Laboratory analyses excluded secondary cognitive disorders, and patients with other ascertained neurological diseases were excluded.

Before performing ^18^F-FDG brain PET, all patients underwent neurological examination, neuropsychological tests (Mini Mental State Examination -MMSE-) and Magnetic Resonance Imaging (MRI), the latter in order to evaluate the morphological brain assessment, especially to detect the presence of atrophy and gliosis as potential signs of white matter chronic cerebral vasculopathy. A final subset of n = 122 cases with confirmed AD (n = 53) or MDI (n = 69) was eventually retained for the study. The standard of reference for the diagnosis was a clinical follow-up of 18–24 months. Table 1 reports demographic, clinical, MMSE and MRI data of the study population; Figure 1 shows the STARD diagram for patient selection. Figure 2 summarizes the whole workflow of the study.

### 2.2. ^18^F-FDG Brain PET/CT

#### 2.2.1. Acquisition Protocol

Before the procedure, written informed consent was obtained from all patients, and their data were treated in accordance with the local privacy rules and regulations. In the informed consent, the patients signed to accept that their data could be used for scientific purposes. The present study was in accordance with the Helsinki Doctrine on Human Experimentation.

^18^F-FDG brain PET/CT was performed according to international guidelines [27]. Patients were advised to fast for at least 4 to 6 h to ensure that ^18^F-FDG uptake would not be influenced by increased serum glucose levels. The latter were checked before injection, and the radiopharmaceutical administration was allowed only if/when the serum glucose values were <160 mg/dl.

The patients were invited to rest comfortably in a quiet, dimly-lit room for at least 15 min before ^18^F-FDG administration and for at least 20 min during the subsequent phase of tracer uptake. They were also instructed not to speak, read, listen to music/sounds or perform any other similar activities during the procedure.

A dose of 3.7 MBq/kg of ^18^F-FDG was then injected intravenously using a previously positioned cannula. Images were acquired 45 min after administration for 15 min at a single bed position using a PET/CT GE Healthcare Discovery 710 tomograph. CT parameters were 120 kVs, 45 effective mAs and one rotation. Slice thickness was 3 mm, and the reconstruction interval was 1.5 mm.

#### 2.2.2. Image Reconstruction and Processing

The iterative reconstruction technique was applied for image reconstruction in axial, coronal and sagittal planes. Three nuclear medicine specialists (SN, AS and PB), all with at least > 20 yr experience in PET neuroimaging, independently interpreted the images. The three specialists were informed about the potentially pathological conditions of the patients but blind to the neurological evaluations.

^18^F-FDG PET/CT images were analyzed both Qualitatively (QL) and Quantitatively (QN). For QL analysis, images were differentiated in normal and pathological metabolism, considering as normal homogenous and symmetric tracer uptake in cortical areas of both hemispheres and as pathological cortical areas with reduced or asymmetric tracer uptake.

QN analysis was performed on a fully-automated post-processing software (Cortex ID SUITE, GE Healthcare, Chicago, IL, United States). All scans, spatially realigned and normalized, were sampled at 16,000 predefined cortical locations and projected on a three-dimensional image. The data were further normalized to the pons and compared with a normal, age-matched segmented database. Finally, a three-dimensional stereotactic surface projection and a Z-score metabolic map were produced [28,29]. In particular, the software computed the radiotracer uptake at 25 predefined regions of interest (ROI), compared the values with those of normal subjects and returned the deviation in terms of Z-score. The ROIs corresponded to the following anatomical cortical areas: prefrontal lateral left (L) and right (R), prefrontal medial L and R, sensorimotor L and R, anterior cingulate L and R, posterior cingulate L and R, precuneus L and R, parietal superior L and R, parietal inferior L and R, occipital lateral L and R, primary visual L and R, temporal lateral L and R, temporal mesial L and R, and whole cerebellum. Z-scores <= −2.0 were considered significant, and for each patient, the maximum negative values achieved in each ROI were also evaluated [19,20].

For univariate analysis and automated classification models, 122 of the 150 initial cases with the diagnosis confirmed by 18–24 months of clinical follow-up (53 AD and 69 MCI) were considered. The remaining 28 patients were excluded from the analysis since the initial clinical suspected diagnosis was not confirmed.

We provide the complete anonymous dataset as Appendix A.

### 2.3. Statistical Analysis

Univariate analysis was performed to determine, for each area, whether there were statistically significant differences in the z scores between the AD and MCI groups. The analysis was based on Welch’s test at a significance level α = 0.05; Bonferroni correction was also applied to counteract the effects of multiple tests. Pairwise correlation between the significant features was assessed via Pearson’s correlation coefficient.

### 2.4. Classification

The automated classification was carried out to estimate the ability of the z scores from the areas with significant differences to discriminate between AD and MCI cases. Three classification models were considered for this task: classification tree (ClT), ridge classifier (RC) and linear Support Vector Machine (lSVM). For each classifier, optimal hyper-parameter values (see Table 2) were determined by grid-search and four-fold cross-validation over 50 random splits; the final accuracy was estimated via leave-one-out cross-validation.

### 2.5. Execution, Data and Code Availability

Data analysis and visualization were based on Python 3.8.4 and functions from the matplotlib, numpy, Pandas, scikit-learn and seaborn packages. The experiments were carried out on an ASUS ProArt Laptop PC with Intel Core^TM^ i7-9750H @ 2.60 GHz CPU, 36 Gb RAM and Windows 10 Pro 64-bit. The total execution time was ≈2 min.

## 3. Results

Univariate analysis (Table 3) returned 14 areas where the z scores were significantly different between the AD and MCI groups. These were: prefrontal lateral (L and R), prefrontal medial (L and R), posterior cingulate (L and R), precuneus (L and R), parietal inferior (L and R), occipital lateral L temporal lateral (L and R) and temporal mesial L. Figure 3 reports the box-plots, strip-plots and *p*-values for each area.

The correlation analysis (Figure 4) identified 19 pairs of areas with very strong positive correlation (Pearson’s *r* ≥ 0.8) [30], 44 with moderately strong positive correlation (0.6 ≤ *r*
*<* 0.8) and 28 with fair positive correlation (0.3 ≤ *r*
*<* 0.6). There were no pairs of areas with poor (0.0 ≤ *r*
*<* 0.3) or negative correlation. The pairs with the strongest correlation (*r* > 0.85) were: posterior cingulate L and R, parietal inferior R and temporal lateral R, parietal inferior L and temporal lateral L, precuneus L and parietal inferior L, prefrontal lateral L and R, parietal inferior L and R, precuneus R and parietal inferior R, prefrontal lateral L and parietal inferior L, precuneus L and R, and prefrontal medial L and R.

As can be seen from Table 4, the classification accuracy ranged between 74.59% (91/122) and 76.23% (93/122), with ClT and RC providing the best results. As for the classification tree, it is worth noting that the best classification strategy consisted of one single split with a cut-off value of ≈ −2.0 on the z score from temporal lateral left area: cases below this threshold were classified as AD and those above the threshold as MCI (Figure 5).

## 4. Discussion

In the Amyloid-PET era, ^18^F-FDG PET/CT still plays a significant role in the diagnosis of Alzheimer Disease (AD) and Mild Cognitive Impairment (MCI) [31,32,33]. In 2011, the National Institute on Aging and Alzheimer’s Association (NIA-AA) proposed separate diagnostic recommendations for the preclinical mild cognitive impairment and dementia stages of Alzheimer’s disease based on different biomarkers able to discriminate, in vivo, the different pathological entities, including beta-amyloid deposition, pathologic tau and neurodegeneration. This classification generated an [A/T/N] system where “A” represents the biomarkers of Ab plaques (including cortical amyloid PET ligand binding or low CSF Ab42), “T” the biomarkers of fibrillary tau (including elevated CSF phosphorylated tau (P-tau) and cortical tau PET ligand binding) and “N” the labeled biomarkers of neurodegeneration or neuronal injury as, CSF T-tau, ^18^FDG-PET/CT hypometabolism and atrophy on MRI [34]. This interesting system represented a basis for a biological definition of AD [6] and paved the way for subsequent studies [35,36,37].

Iaccarino [38] showed that in a group of 518 MCI subjects and 269 healthy controls from the Alzheimer’s Disease Neuroimaging Initiative (ADNI) database (adni.loni.usc.edu), a positive amyloid PET scan was not associated with clinical progression in the majority (≥60%) of subjects, although it represented a significant risk factor, while a negative ^18^F-FDG PET/CT scan at baseline strongly predicted clinical stability with high negative predictive values (>0.80) for both groups of subjects. The authors concluded that ^18^F-FDG PET/CT brain metabolism or other neurodegeneration measures should be coupled with amyloid-PET to identify clinically stable individuals in order to exclude them from clinical trials.

Ottoy and co-workers [14] showed that patients with MCI progressed to AD at an annual rate of 31%, and this could be best predicted by combining neuropsychological testing with MRI-based Hippocampal Volume and ^18^F-FDG PET/CT (specificity = 96%, sensitivity = 92%).

Chatelat et al. [39] reported that ^18^F-FDG PET/CT could predict the clinical outcome in patients with MCI who already have an amyloid-PET scan. In their work, a normal ^18^F-FDG-PET scan was associated with long-term clinical stability–even in amyloid-positive cases; by contrast, a pathological ^18^F-FDG-PET scan was indicative of an increased risk of progressive cognitive decline even in amyloid-negative cases.

Tondo et al. [40] observed 142 subjects with amnestic MCI for 4–19 years and determined that hypometabolism patterns on baseline ^18^F-FDG PET/CT could predict long-term outcomes in terms of stability or progression to AD. Specifically, they reported that limbic-predominant hypometabolism pattern was associated with clinical stability, thus making progression to AD very unlikely.

Arbizu et al. [32] investigated the additional value of ^18^F-FDG PET/CT beyond clinical neuropsychological examination to support the diagnosis of prodromal Alzheimer’s Disease (AD), frontotemporal lobar degeneration (FTLD) and prodromal dementia with Lewy bodies (DLB) in mild cognitive impairment (MCI) subjects. A panel of seven experts (four from the European Association of Nuclear Medicine and 3 from the European Academy) provided recommendations about the incremental value of ^18^F-FDG PET/CT to evaluate the etiology of MCI (AD, FTLD or DLB). The study identified 55 relevant papers, which were obtained through a population, intervention, comparison and outcome (PICO) search string. The meta-analysis indicated that ^18^F-FDG PET/CT patterns enabled the correct identification of MCI etiology due to AD with accuracy between 58% and 100% and area-under-curve between 0.66 and 0.97; however, no specific data were found in regard to MCI due to FTLD or DLB.

However, the clinical use of ^18^F-FDG PET/CT reached the consensus recommendations in MCI subjects for the high negative predictive value and the existence of different disease-specific patterns of hypometabolism. Worthy of mention is that 123I-ioflupane and 123I-MIBG can be useful in the case of prodromal DLB [41].

In this scenario, our findings confirm the ability of ^18^F-FDG PET/CT to differentiate AD from MCI. Specifically, the univariate analysis (Table 1, Figure 3) indicates that AD and MCI cases had significantly different readings in many strategic areas, in particular, precuneus, posterior cingulate and parietal and temporal regions.

We also demonstrated the ability of automated classifiers (ClT, lSVM and RC) to discriminate between AD and MCI based on the z scores of the significant areas. Our classification accuracy ranged between 74.59% (91/122) and 76.23% (93/122), with ClT and RC providing the best results. As for the classification tree, it is worth noting that the best classification strategy consisted of one single split with a cut-off value of ≈ −2.0 on the z score from temporal lateral left area: cases below this threshold were classified as AD and those above the threshold as MCI (Figure 5). This suggests that radiopharmaceutical uptake in the temporal lateral left area is the “marker” region able to discriminate AD and MCI in our series of the patient. This is consistent with the hierarchical pathologic progression of neurofibrillary tangles that spread from the middle temporal to the lateral temporal areas. Indeed, tau pathology initially affects transentorhinal, followed by entorhinal, after fusiform and lingual and later reach lateral temporal association areas [42,43,44].

Finally, we wish to underline the usefulness of semi-quantitative analysis to assist visual reading, as it has been widely evidenced by international literature [15,37]. We used Cortex ID suite, a fully automated post-processing software for quantifying ^18^F-FDG PET/CT and beta-amyloid brain scans that used three-dimensional stereotactic surface projections (3D-SSP) for statistical image analysis [45,46]. Future works may focus on the combination of semi-quantitative analysis with direct extraction of traditional and/or deep learning imaging features from the brain scans, as discussed in [47,48,49].

In conclusion, our study indicates that the combination of semi-quantitative analysis and automatic classification can improve the diagnostic process through the identification of the metabolically impaired areas specific to the different disorders. The development of computer-aided diagnosis (CAD) systems is also receiving attention not only as a means to support the diagnostic process by the calculation of cut-off values but also to assess correlations between clinical data and pathologies. This improves traditional image interpretation and diagnostic assessment in many neurodegenerative diseases [49,50,51]. The role played by artificial intelligence techniques, i.e., Machine Learning, Radiomics and Deep Learning is pivotal to building diagnostic models for personalized care. This is of particular importance in neurodegenerative diseases such as AD and MCI, as they fall in a sort of “grey area” where a clear diagnosis is often difficult. Our paper confirms the clinical value of ^18^F-FDG brain PET/CT as an essential diagnostic first step to contribute to the differential diagnosis of dementia disorders also in the amyloid PET and biological markers (i.e., amyloid and tau protein) era since, as it is well known, accurate and early diagnosis of Alzheimer disease in respect of Mild Cognitive Impairment is crucial for improving the condition of patients. The combined use of semi-quantitative analysis of ^18^F-FDG PET and automatic classification seems to be a supportive tool for clinical diagnosis in order to consider effective preventive early measures to delay the appearance of the full-blown disease or to suggest further investigations such as amyloid PET or more advanced treatments and therapeutic approaches [52,53].

### Limitations and Future Work

A number of limitations apply to this paper, among which are the retrospective nature of the study and the relatively contained sample size. Furthermore, it is to be noted that our method relies on semi/quantitative data, the calculation of which is delegated to an external software package (Cortex ID). Extraction of custom imaging features directly from the PET/CT scans via hand-crafted methods and/or Deep Learning (as discussed in [54,55]) is an interesting subject for future studies.

## Figures and Tables

**Figure 1 diagnostics-12-02425-f001:**
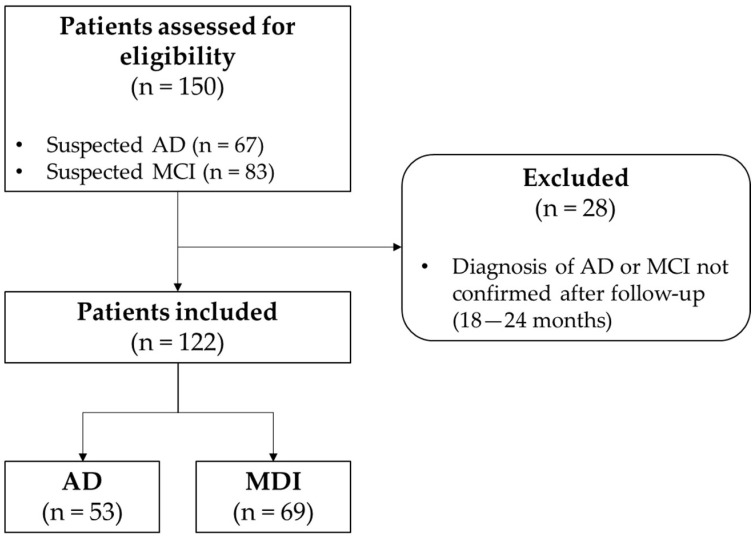
STARD diagram for the study population.

**Figure 2 diagnostics-12-02425-f002:**
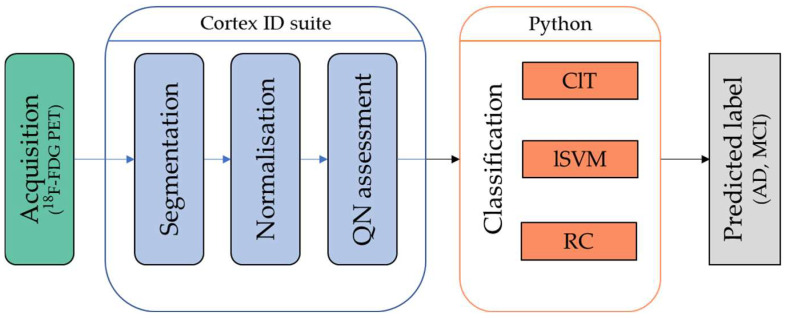
Overall procedure workflow.

**Figure 3 diagnostics-12-02425-f003:**
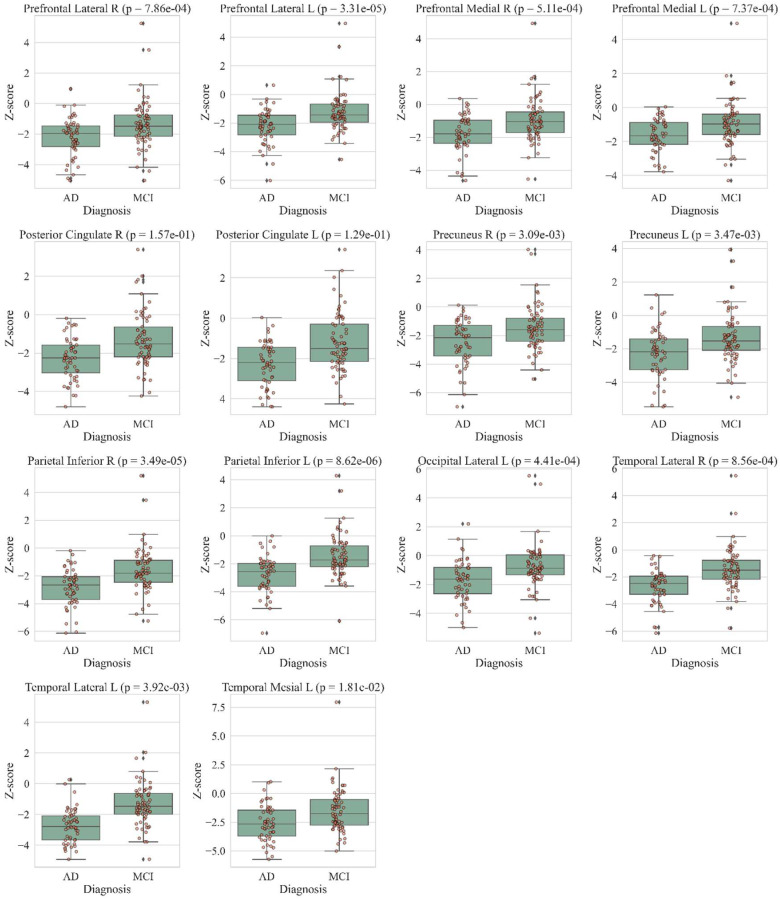
Box-plots and strip-plots of the Z-score for each area with significant difference between AD and MCI.

**Figure 4 diagnostics-12-02425-f004:**
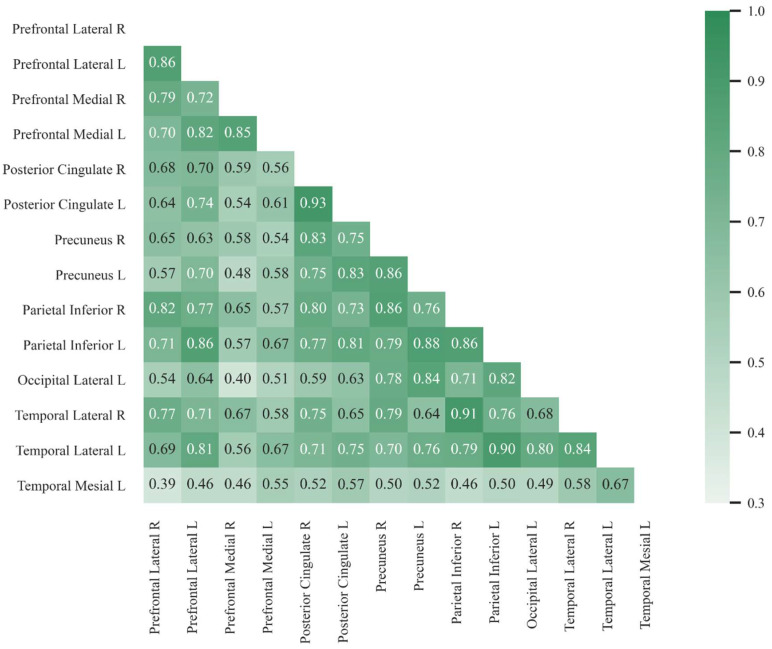
Correlation heat-map. Values report Pearson’s correlation coefficient (*r*) between the z scores from each pair of areas.

**Figure 5 diagnostics-12-02425-f005:**
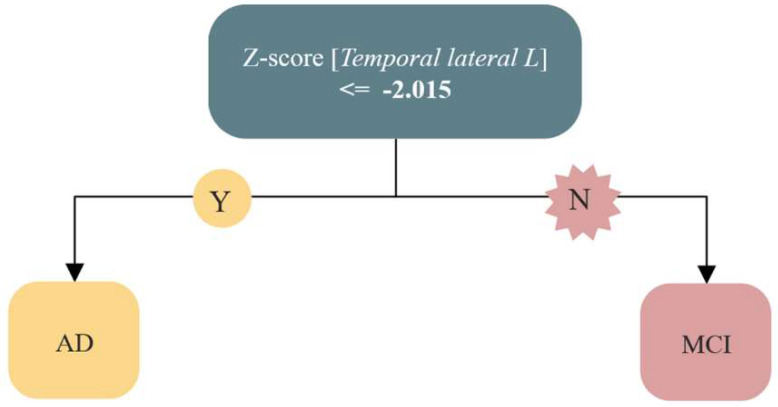
Prototype classification tree. Key to abbreviations: AD = Alzheimer’s disease, MCI = Mild Cognitive Impairment.

**Table 1 diagnostics-12-02425-t001:** Demographic, clinical, Mini Mental State Examination (MMSE) and Magnetic Resonance Imaging (MRI) data of Alzheimer Disease (AD) and Mild Cognitive Impairment (MCI) patients.

**AD** 67 cases			**MCI** 83 cases	
**Age**	Range 55–83 year	Mean ±standard deviation 69.5 ± 8.64	Range 40–85 year	Mean ±standard deviation 71.4 ± 9.37
**Sex**	29 male	38 female	35 male	48 female
**Family history for dementia**	Positive 25/67	Negative 42/67	Positive 38/83	Negative 45/83
**Correct Mini Mental State Examination (MMSE)**	Range 9.9/30–26/30	Mean ±standard deviation 22 ± 4.8	Range 25.3/30–30/30	Mean ±standard deviation 25.3 ± 3.04
**MRI**	Slight to severe atrophy: 15/67 cases		Slight to severe atrophy: 17/83 cases	
	Focal/diffuse gliosis: 15/67 cases		Focal/diffuse gliosis: 16/83 cases	
	Diffuse cerebrovascular lesions and atrophy: 19/67 cases		Diffuse cerebrovascular lesions and atrophy: 20/83 cases	
	No significant alteration: 18/67 cases		No significant alteration: 30/83 cases	

**Table 2 diagnostics-12-02425-t002:** Hyper-parameter tuning. Key to symbols: SC = splitting criterion; MD = maximum depth of the classification tree; *C* = regularization parameter; *α* = regularization strength.

Classification Model	Grid Search Domain	Optimal Hyper-Parameters
Classification tree	SC = {“entropy”, “gini”} MD = {1, 2, 4, 6, 8}	SC = “gini” MD = 1
Linear SVM	*C* = {0.01, 0.1, 1.0, 10.0}	*C* = 1.0
Ridge classifier	*α* = {0.01, 0.1, 1.0, 10.0}	*α* = 1.0

**Table 3 diagnostics-12-02425-t003:** Results of the univariate analysis. The values report the corresponding z score (mean ± std) for each area. Key to abbreviations: AD = Alzheimer disease, MCI = Mild Cognitive Impairment.

Area	Diagnosis		*p*-Value	Significant
	AD	MCI		
Prefrontal Lateral R	−2.21 ± 1.23	−1.33 ± 1.54	<0.001	Yes
Prefrontal Lateral L	−2.24 ± 1.18	−1.24 ± 1.38	<0.001	Yes
Prefrontal Medial R	−1.73 ± 1.12	−0.94 ± 1.30	<0.001	Yes
Prefrontal Medial L	−1.64 ± 0.96	−0.92 ± 1.31	<0.001	Yes
Sensorimotor R	−0.93 ± 1.52	−0.53 ± 1.51	0.157	No
Sensorimotor L	−0.89 ± 1.37	−0.49 ± 1.48	0.129	No
Anterior Cingulate R	−1.30 ± 0.87	−0.74 ± 1.17	0.003	No
Anterior Cingulate L	−1.27 ± 0.82	−0.72 ± 1.22	0.003	No
Posterior Cingulate R	−2.29 ± 1.08	−1.31 ± 1.41	<0.001	Yes
Posterior Cingulate L	−2.28 ± 1.10	−1.21 ± 1.42	<0.001	Yes
Precuneus R	−2.48 ± 1.56	−1.44 ± 1.55	<0.001	Yes
Precuneus L	−2.27 ± 1.43	−1.35 ± 1.47	<0.001	Yes
Parietal Superior R	−2.15 ± 1.35	−1.38 ± 1.54	0.004	No
Parietal Superior L	−1.76 ± 1.37	−1.11 ± 1.60	0.018	No
Parietal Inferior R	−2.84 ± 1.32	−1.56 ± 1.56	<0.001	Yes
Parietal Inferior L	−2.76 ± 1.30	−1.46 ± 1.51	<0.001	Yes
Occipital Lateral R	−1.47 ± 1.50	−0.65 ± 1.48	0.004	No
Occipital Lateral L	−1.70 ± 1.39	−0.75 ± 1.54	<0.001	Yes
Primary Visual R	−0.69 ± 1.27	−0.41 ± 1.27	0.225	No
Primary Visual L	−0.68 ± 1.19	−0.32 ± 1.36	0.128	No
Temporal Lateral R	−2.73 ± 1.22	−1.41 ± 1.55	<0.001	Yes
Temporal Lateral L	−2.74 ± 1.10	−1.31 ± 1.46	<0.001	Yes
Temporal Mesial R	−2.13 ± 1.43	−1.36 ± 1.89	0.012	No
Temporal Mesial L	−2.56 ± 1.56	−1.46 ± 1.90	<0.001	Yes
Cerebellum Whole	−0.49 ± 1.30	−0.37 ± 1.26	0.613	No

**Table 4 diagnostics-12-02425-t004:** Classification performance. Accuracy was estimated via leave-one-out cross-validation.

Classification Model	Accuracy
Classification tree	76.23% (93/122)
Linear SVM	76.23% (93/122)
Ridge classifier	74.59% (91/122)

## Data Availability

The supporting data can be found in the Appendix A.

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
