# Peer review of "Differential Diagnosis of Alzheimer Disease vs. Mild Cognitive Impairment Based on Left Temporal Lateral Lobe Hypomethabolism on 18F-FDG PET/CT and Automated Classifiers"

_diagnostics, 2022, doi:10.3390/diagnostics12102425_

Round 1

Reviewer 1 Report

A focus is stated in the opening line of the study: examine the capacity of Artificial Intelligence with automatic categorization for Alzheimer's, although the paper does not address this portion. There are no technical information, such as what architectures were used or how they were implemented. The materials and techniques section mentions classification trees, ridge classifiers, and support vector machines, none of which are novel. According to the opinion of this reviewer, this work does not give any originality or provide merit for publishing, nor does it provide state-of-the-art information (which falls in the category of deep learning, not investigated in this paper). The categorization findings are low in accuracy and far inferior to those of state-of-the-art studies.

Author Response

Dear reviewer thanks a lot for your useful comments. You will find below the answers to your remarks point by point.

Remark 1.1: A focus is stated in the opening line of the study: examine the capacity of Artificial Intelligence with automatic categorization for Alzheimer's, although the paper does not address this portion.

Authors’ reply: We are a bit surprised at the reviewer’s remark as the abstract clearly states that the purpose of this work is to improve the differential diagnosis between Alzheimer Disease (AD) and Mild Cognitive Impairment (MCI) based on semi-quantitative data from brain 18F-FDG PET/CT. This is indeed the main focus of this study.

Remark 1.2: There are no technical information, such as what architectures were used or how they were implemented.

Authors’ reply: Details about the machine learning methods used, the procedure for hyperparameter tuning and the implementation platform are all provided in the Materials and Methods section (subsections Classification, and Data and Code Availability).

Remark 1.3: The materials and techniques section mentions classification trees, ridge classifiers, and support vector machines, none of which are novel.

Authors’ reply: We agree with the reviewer that the machine learning methods used in this work are fairly standard, but we also believe this is the right choice here. Indeed the main objective of this work is not the development of new machine learning methods – that would be outside the scope of this study and possibly of the journal too – but again to examine the ability of existing methods to discriminate between AD and MCI on semi-quantitative data from brain 18F-FDG PET/CT.

Remark 1.4: According to the opinion of this reviewer, this work does not give any originality or provide merit for publishing, nor does it provide state-of-the-art information (which falls in the category of deep learning, not investigated in this paper).

Authors’ reply: We agree with the reviewer on the importance of Deep Learning (DL) in general; however, this is not the ideal context of application for DL, since we operate on semi-quantitative data generated by neuroimage analysis software (CORTEX ID SUITE, GE Healthcare). The use of DL directly on the 18F-FDG PET/CT scan could be interesting subject for future studies.

Remark 1.5: The categorization findings are low in accuracy and far inferior to those of state-of-the-art studies.

Authors’ reply: The categorization accuracy achieved in this work seem consistent with the degree of overlap between the two classes (AD and MCI) as shown in the box-plots/strip-plots. As for the accuracy values, although comparing studies with different study populations is always risky, we observe that in absolute terms our results are in good agreement with the literature – and for instance well within the range (58–100%) reported in the review paper by Arzibu et al. [33].

Reviewer 2 Report

I am delighted to review the article entitled "DIFFERENTIAL DIAGNOSIS OF ALZHEIMER DISEASE VS 2 MILD COGNITIVE IMPAIRMENT BASED ON LEFT TEM- 3 PORAL LATERAL LOBE HYPOMETHABOLISM ON 18F-FDG 4 PET/CT AND AUTOMATED CLASSIFIERS "

The author proposed three automated classification models on brain 18F-FDG PET/CT images in differentiating AD from MCI. 

Combining semi-quantitative analysis and automatic classification improved the diagnostic process.

Some modifications are requested to improve the quality of the study:

- Please indicate the inclusion/exclusion criteria. A STARD flow diagram may help the reader;

- The segmentation process is not clear. It was a manual definition of the ROI or an automatic process using an existing template?

- The authors should produce a Figure of the workflow of all methods adopted (segmentation, normalization, QN assessment, SVM used, patients stratification)

- In the discussion, the authors should discuss other recent studies on the application of radiomics analysis of brain [ 18 F]FDG PET/CT to predict beta-amyloid positivity and Alzheimer's Disease;

- Please discuss if the methods adopted are feasible and reproducible in clinical reporting;

- The accuracy of FDG-PET in the differential diagnosis of AD is already noted in the literature. Any added value of the method adopted?

- The limitation of the study has not been stated.

Author Response

Dear reviewer thanks a lot for your useful considerations. You will find below the answers to your remarks point by point.

Remark 2.1: Please indicate the inclusion/exclusion criteria. A STARD flow diagram may help the reader.

Authors’ reply: We have added a STARD diagram to clarify the inclusion/exclusion criteria.

Remark 2.2: The segmentation process is not clear. It was a manual definition of the ROI or an automatic process using an existing template?

Authors’ reply: Segmentation for QN analysis was performed on a fully-automated post processing Software (Cortex ID SUITE, GE Healthcare, Chicago, IL, United States). All scans, spatially realigned and normalized, were sampled at 16,000 predefined cortical locations and projected on a three-dimensional image. The data were further normalized to the pons and compared with normal, age-matched segmented database. Finally, a three-dimensional stereotactic surface projection and a Z-score metabolic map were produced [29, 30]. In particular, the software computed the radiotracer uptake at 25 predefined regions of interest (ROI), compared the values with those of normal subjects and returned the deviation in terms of Z-score. The ROIs corresponded to the following anatomical cortical areas: prefrontal lateral left (L) and right (R), prefrontal medial L and R, sensorimotor L and R, anterior cingulate L and R, posterior cingulate L and R, precuneus L and R, parietal superior L and R, parietal inferior L and R, occipital lateral L and R, primary visual L and R, temporal lateral L and R, temporal mesial L and R, and whole cerebellum. Z-scores <= -2.0 were considered significant and, for each patient, the maximum negative values achieved in each ROI were also evaluated.

We have updated the Materials and Methods section accordingly.

Remark 2.3: The authors should produce a Figure of the workflow of all methods adopted (segmentation, normalization, QN assessment, SVM used, patients stratification)

Authors’ reply: We have added a figure illustrating the whole workflow.

Remark 2.4: In the discussion, the authors should discuss other recent studies on the application of radiomics analysis of brain [18 F]FDG PET/CT to predict beta-amyloid positivity and Alzheimer's Disease

Authors’ reply: We have updated the discussion with references to the following recent studies.

  • Alongi P., et al. Radiomics Analysis of Brain [18 F]FDG PET/CT to Predict Alzheimer’s Disease in Patients with Amyloid PET Positivity: A Preliminary Report on the Application of SPM Cortical Segmentation, Pyradiomics and Machine-Learning Analysis
    (2022) Diagnostics, 12 (4), art. no. 933
  • Zhou P. et al Deep-Learning Radiomics for Discrimination Conversion of Alzheimer's Disease in Patients With Mild Cognitive Impairment: A Study Based on 18F-FDG PET Imaging
    (2021) Frontiers in Aging Neuroscience, 13, art. no. 764872
  • Cui W. et al. BMNet: A New Region-Based Metric Learning Method for Early Alzheimer’s Disease Identification With FDG-PET Images (2022) Frontiers in Neuroscience, 16, art. no. 831533

Remark 2.5: Please discuss if the methods adopted are feasible and reproducible in clinical reporting

Authors’ reply: The overall aim of this study was to demonstrate the ability of machine learning methods to discriminate between AD and MCI from semi-quantitative analysis of FDG PET/CT images of the brain. The proposed method is relatively uncomplicated and easy to implement, therefore in principle suitable for clinical reporting. Our results are also encouraging; however, they should be confirmed in larger, ideally perspective studies before translation into the clinical practice.

Remark 2.6: The accuracy of FDG-PET in the differential diagnosis of AD is already noted in the literature. Any added value of the method adopted?

Authors’ reply: We agree with the reviewer that accuracy of FDG-PET in the differential diagnosis of AD is already noted in the literature; still, our study suggests that the combination of semi-quantitative analysis and artificial intelligence methods (automatic classifiers) can improve the diagnostic process – in particular through the identification of the metabolically impaired areas that are significant to differentiate the disorders.

Remark 2.7: The limitation of the study has not been stated

Authors’ reply: A number of limitations apply to this paper, among which the retrospective nature of the study and the relatively contained sample size. Furthermore, it is to be noted that our method relies on semi/quantitative data the calculation of which is delegated to an external software package (Cortex ID). Extraction of custom imaging features directly from the PET/CT scans via hand-crafted methods and/or Deep Learning is interesting subject for future studies.

We have reported the above considerations in the Limitations and future work section of the revised version of the manuscript.  

Reviewer 3 Report

The manuscript presents the evaluation of Artificial Intelligence (AI) with automatic classification methods applied to semi-quantitative data from brain 18F-FDG PET/CT to improve the differential diagnosis between Alzheimer Disease (AD) and Mild Cognitive Impairment (MCI). In doing so, the authors retrospectively analyzed a total of 150 consecutive patients who underwent diagnostic evaluation for suspected AD (n = 67) or MCI (n = 83). As conclusion, the authors confirm the usefulness of brain 18F-FDG PET/CT QL and QN analyses in differentiating AD from MCI. In summary, the paper looks valid and fits within the scope of Diagnostics. The references should be checked, there seem some formation errors.

Author Response

Dear reviewer thanks a lot for your useful considerations. You will find below the answers to your remarks point by point.

Remark 3.1: The manuscript presents the evaluation of Artificial Intelligence (AI) with automatic classification methods applied to semi-quantitative data from brain 18F-FDG PET/CT to improve the differential diagnosis between Alzheimer Disease (AD) and Mild Cognitive Impairment (MCI). In doing so, the authors retrospectively analyzed a total of 150 consecutive patients who underwent diagnostic evaluation for suspected AD (n = 67) or MCI (n = 83). As conclusion, the authors confirm the usefulness of brain 18F-FDG PET/CT QL and QN analyses in differentiating AD from MCI. In summary, the paper looks valid and fits within the scope of Diagnostics.

Authors’ reply: We are glad to receive an overall positive evaluation of the work.

Remark 3.2: The references should be checked, there seem some formation errors.

Authors’ reply: We have double checked the references to correct errors and inconsistencies.

Round 2

Reviewer 1 Report

The article focuses on application rather than basic innovation. Rebuttal essay fails to demonstrate the value of simple machine learning versus efficiently training a specific architecture. The authors did not create the operational concept. The literature evaluation is poorly constructed, and the authors miss similar models such as DOI 10.3390/s22030740, 10.1002/int.22856, and others. Professors Modupe, Li, Mandarin, Damashewich, and Bei Shan are well-known in this subject for their work in artificial intelligence and medicine. The experimental part is absent, which is unfortunate considering the journal's goal. The article lacks a competent medical review of the outcomes, and the data is statistically inconclusive, and therefore cannot be treated as proof or a real claim that the application works and is beneficial, especially given the relatively restricted set of MCI, and even further considering only two classes are in question (other researchers try to differentiate within a minimum of 4). Finally, there is no mention of computing performance (FLOPS) when basic methods are used (supposedly a speedy benefit...). The training is not described in full in the article. There is no reliable proof of non-overfitting. There is no comparison to current work that has produced greater accuracy, for example in ADNI:

https://doi.org/10.3390/s22113966

https://doi.org/10.1007/s13042-021-01501-7

https://doi.org/10.1002/jemt.23861

https://doi.org/10.1109/JBHI.2021.3097721

https://doi.org/10.3390/diagnostics11061071

https://doi.org/10.1016/j.mri.2021.02.001

https://doi.org/10.1016/j.neucom.2020.09.012

https://doi.org/10.3390/s22030740

https://doi.org/10.1186/s13195-021-00797-5

https://doi.org/10.1093/bib/bbac022

https://doi.org/10.1038/s41598-022-06444-9

https://doi.org/10.1007/s11042-022-12228-0

Author Response

dear prof.

according to your suggestions we provided to revise the manuscript 

Remark 1.1: The article focuses on application rather than basic innovation. Rebuttal essay fails to demonstrate the value of simple machine learning versus efficiently training a specific architecture.

Authors’ reply: We are surprised to hear the reviewer is not entirely satisfied with our reply. As we stated in the paper, we would like to emphasize once more that it is not the aim of this work to compare machine learning vs training a specific architecture for the task investigated here. The reason is that our procedure works on semi-quantitative data generated by dedicated software, which is not the ideal context of application for deep learning. For this reason Deep Learning is outside the scope of this work. However, we acknowledge that the use of DL directly on the 18F-FDG PET/CT scans could be interesting subject for future studies (please see Limitations and future studies).

Remark 1.2: The authors did not create the operational concept

Authors’ reply: We believe that the operational concept is described in detail the Materials and Methods section and clearly summarised in Fig. 2.

Remark 1.3: The literature evaluation is poorly constructed, and the authors miss similar models such as DOI 10.3390/s22030740, 10.1002/int.22856, and others. Professors Modupe, Li, Mandarin, Damashewich, and Bei Shan are well-known in this subject for their work in artificial intelligence and medicine

Authors’ reply: We thank the reviewer for suggesting the two references above, which are now cited in the manuscript. We would like to emphasize, however, that both references focus on Deep Learning – hence they are not directly related to the method presented here. However, they are certainly relevant for future studies (please see also our reply to remark 1.1. on this point).

Remark 1.4: The experimental part is absent, which is unfortunate considering the journal's goal.

Authors’ reply: There is indeed an experimental part, which consists of univariate analysis, correlation analysis and automatic classification. This is all described in Materials and Methods and related subsections.

Remark 1.5: The article lacks a competent medical review of the outcomes, and the data is statistically inconclusive, and therefore cannot be treated as proof or a real claim that the application works and is beneficial, especially given the relatively restricted set of MCI, and even further considering only two classes are in question (other researchers try to differentiate within a minimum of 4).

Authors’ reply: As the reviewer may observe in our conclusion, we underlined that our study indicates that the combination of semi-quantitative analysis and automatic classification can improve the diagnostic process through the identification of the metabolically impaired areas specific to the different disorders. The development of computer aided diagnosis (CAD) systems is also receiving attention not only as a means to support the diagnostic process by the calculation of cut-off values; but also to assess correlations between clinical data and pathologies. This improves on traditional image interpretation and diagnostic assessment in many neurodegenerative diseases [50, 51]. The role played by artificial intelligence techniques, i.e. Machine Learning, Radiomics and Deep Learning, is pivotal to building diagnostic models for personalized care. This is of particular importance in neurodegenerative diseases such as AD and MCI, as they fall in a sort of ‘grey area’ where a clear diagnosis in often difficult. Our paper confirms the clinical value of 18F-FDG brain PET/CT as an essential diagnostic first step to contribute to the differential diagnosis of dementia disorders also in the amyloid PET and biological markers (i.e. amyloid and tau protein) era.  However, in accordance with the suggestion we had better explained this part adding some new references also listed below.

Huang Z, Sun M, Guo C. Automatic Diagnosis of Alzheimer's Disease and Mild Cognitive Impairment Based on CNN + SVM Networks with End-to-End Training. Comput Intell Neurosci. 2021 Aug 13;2021:9121770.

Doroszkiewicz J, Mroczko B. New Possibilities in the Therapeutic Approach to Alzheimer's Disease. Int J Mol Sci. 2022 Aug 10;23(16):8902.

Remark 1.6: Finally, there is no mention of computing performance (FLOPS) when basic methods are used (supposedly a speedy benefit...).

Authors’ reply: The proposed method operates on semi-quantitative data and is therefore particularly quick, as the reviewer correctly noted. We have updated the Materials and Methods/Execution, Data and Code Availability section with details about the hardware used and execution time.

Remark 1.7: The training is not described in full in the article. There is no reliable proof of non-overfitting.

Authors’ reply: Details about train/test classification procedure and hyperparameters search are available in Materials and Methods/Classification

Remark 1.8: There is no comparison to current work that has produced greater accuracy, for example in ADNI

Authors’ reply: We are surprised at the reviewer’s remark at this stage of the revision process. The manuscript has already been revised in that sense – in particular the Discussion section.